# Colorimetric and Electrochemical Methods for the Detection of SARS-CoV-2 Main Protease by Peptide-Triggered Assembly of Gold Nanoparticles

**DOI:** 10.3390/molecules27030615

**Published:** 2022-01-18

**Authors:** Yunxiao Feng, Gang Liu, Ming La, Lin Liu

**Affiliations:** 1College of Chemistry and Chemical Engineering, Pingdingshan University, Pingdingshan 467000, China; 2743@pdsu.edu.cn; 2Henan Province of Key Laboratory of New Optoelectronic Functional Materials, College of Chemistry and Chemical Engineering, Anyang Normal University, Anyang 455000, China; liugang08215@163.com; 3College of Chemistry and Chemical Engineering, Henan University of Technology, Zhengzhou 450011, China

**Keywords:** SARS-CoV-2 main protease, colorimetry, electrochemical impedance spectroscopy, gold nanoparticles

## Abstract

Severe acute respiratory syndrome coronavirus 2 (SARS-CoV-2) main protease (Mpro) has been regarded as one of the ideal targets for the development of antiviral drugs. The currently used methods for the probing of Mpro activity and the screening of its inhibitors require the use of a double-labeled peptide substrate. In this work, we suggested that the label-free peptide substrate could induce the aggregation of AuNPs through the electrostatic interactions, and the cleavage of the peptide by the Mpro inhibited the aggregation of AuNPs. This fact allowed for the visual analysis of Mpro activity by observing the color change of the AuNPs suspension. Furthermore, the co-assembly of AuNPs and peptide was achieved on the peptide-covered electrode surface. Cleavage of the peptide substrate by the Mpro limited the formation of AuNPs/peptide assembles, thus allowing for the development of a simple and sensitive electrochemical method for Mpro detection in serum samples. The change of the electrochemical signal was easily monitored by electrochemical impedance spectroscopy (EIS). The detection limits of the colorimetric and electrochemical methods are 10 and 0.1 pM, respectively. This work should be valuable for the development of effective antiviral drugs and the design of novel optical and electrical biosensors.

## 1. Introduction

The outbreak of pneumonia caused by the new coronavirus started from the beginning of 2020 in a global pandemic. The new coronavirus was defined as severe acute respiratory syndrome coronavirus 2 (SARS-CoV-2) by the International Committee on Taxonomy of Viruses and the pneumonia was named coronavirus disease 2019 (2019-nCoV) by the World Health Organization (WHO). The currently approved drugs for treating the infected patients show limited and toxic side effects. Thus, more effective antiviral drugs are still urgently desired [1,2,3]. The SARS-CoV-2 main protease (Mpro), also known as 3C-like protease (3CLpro), is involved in cleaving the viral polyprotein to produce the essential viral protein required for virus replication and pathogenesis. The protease plays a leading physiological role in the life cycle of the virus and has been regarded as one of the ideal targets for the design of antiviral drugs [4,5,6,7,8]. The commercial kits for the detection of the Mpro and the screening of its inhibitors mainly adopt the fluorescence resonance energy transfer (FRET) method, in which the Mpro catalyzes the hydrolysis of a double-labeled peptide substrate with an acceptor and donor couple [9,10]. Although the approach is sensitive and rapid, it requires the use of an expensive and complicated peptide substrate and advanced instruments [11]. For this reason, it is of importance to develop simple, sensitive, cost-efficient and high-throughput methods for the rapid detection of the Mpro and the screening of its potential inhibitors.

Gold nanoparticles (AuNPs) exhibit a high extinction coefficient and unique size-dependent optical properties. AuNPs-based colorimetric methods have been widely used in the field of biological analysis because of their simple sample processing and low instrument investment [12,13,14,15]. Peptides with cysteine and/or positively charged amino acid (e.g., Lys and Arg) residues can induce the aggregation of AuNPs through the Au-S and electrostatic interactions [16,17,18,19,20,21,22]. The protease-catalyzed cleavage of the peptide may regulate the aggregation of AuNPs, thus allowing for the detection of protease activity and the screening of its inhibitors. In the commercial kits, a double-labeled peptide with a sequence of acceptor–KTSAVLQSGFRKME–donor can be used as the substrate to monitor Mpro activity. When the substrate is cleaved into two short segments by the Mpro, the acceptor and donor labeled at the two ends of the peptide are separated, resulting in the fluorescence recovery of the donor. In the present work, we found that the label-free peptide with a sequence of RKTSAVLQSGFRK can induce the aggregation of AuNPs through the electrostatic interactions (Figure 1A). Cleavage of the peptide by the Mpro inhibits the aggregation of AuNPs, thus allowing for the visual analysis of the Mpro by monitoring the color change of the AuNPs suspension. The AuNPs-based homogeneous method is simple, rapid, cost-efficient and high-throughput for monitoring Mpro activity and the screening of its inhibitors, but both the sensitivity and the anti-interference ability of the method for determining the Mpro in biological samples are poor. Electrochemical biosensors can sensitively monitor the cleavage event of the peptide at the liquid–solid interface. Recent studies have shown that the aggregation of nanoparticles induced by the targets can be initiated on the electrode surface, thus implanting one principle into another field [22,23,24,25,26,27,28,29]. Herein, we propose that the liquid phase analysis of the Mpro can be transformed into an electrochemical analysis by regulating the peptide-triggered assembly of AuNPs on the peptide-covered electrode surface (Figure 1B). The resulting peptide/AuNPs networks can promote the electron transfer of [Fe(CN)_6_]^3−/4−^ due to the excellent conductivity and large specific surface area of AuNPs and the positive charges of the peptide framework [22,30]. However, when the peptide substrate immobilized on the electrode surface was cleaved by the Mpro, the assembly of AuNPs on the electrode surface would be prevented. The change of electron transfer resistance can be easily monitored by electrochemical impedance spectroscopy (EIS). The electrochemical method integrates the advantages of colorimetric analysis and surface-tethered biosensors. Finally, the colorimetric method was used to evaluate the activity of the Mpro in the absence and presence of inhibitors, and the electrochemical strategy was employed to determine the Mpro in biological samples.

## 2. Results and Discussion

### 2.1. Feasibility for Colorimetric Analysis of Mpro

For the colorimetric analysis of Mpro activity, a label-free peptide substrate (RKTSAVLQSGFRK) was designed according to the sequence of the commercial peptide substrate (acceptor–KTSAVLQSGFRKME–donor). The positively charged Lys and Arg residues at both ends can bind with AuNPs through the electrostatic interactions, thus leading to the aggregation of AuNPs. As shown in Figure 1A, the surface plasmon resonance bond of AuNPs changed from 520 nm to 585 nm with the addition of the peptide substrate (curves 1 and 2), which is accompanied by the change of the solution color from red to blue (tubes 1 and 2). The result demonstrated that the peptide can induce the aggregation of AuNPs, which is also confirmed by DLS analysis. Interestingly, when the peptide substrate was incubated with the Mpro for a given time, no apparent changes in the solution color and UV-Vis spectrum were observed (tube/curve 3), demonstrating that the cleavage of the peptide by the Mpro prevented the aggregation of AuNPs. Thus, the activity and concentration of the Mpro may be easily monitored by discriminating the solution color with the naked eye in a convenient and straightforward way. The peptide concentration is crucial for the aggregation of AuNPs. We found that the changes of the solution color and the UV-Vis spectrum were dependent upon the concentration of the peptide. The ratio of the absorbance at 585 nm and 520 nm (A_585_/A_520_) was used to evaluate the degree of AuNPs aggregates. The A_585_/A_520_ was intensified with the increase of the peptide concentration in the range of 0.1~7.5 μM. Since a high concentration of the peptide substrate may decrease the sensitivity of the method, a compromised concentration of the peptide (5 μM) was used for the detection of the Mpro in the follow-up quantitative analysis.

### 2.2. Sensitivity for Colorimetric Analysis of Mpro

The sensitivity of the colorimetric method was evaluated by measuring different concentrations of the Mpro. As shown in Figure 2, the solution color gradually became purple and red from blue with the increase of the Mpro concentration, which was accompanied by the decrease of the absorption intensity at 585 nm. The A_585_/A_520_ value decreased linearly with the Mpro concentration change from 0.01 to 0.5 nM. The linear equation can be expressed as A_585_/A_520_ = 0.93–0.85 [Mpro] (nM). The lowest detectable concentration was comparable to that measured by the commercial fluorescence kit. However, the colorimetric method can be performed in a rapid and straightforward way and the peptide substrate is cheap and stable for long-term storage. To demonstrate the application of the method, the inhibitory effect of ebselen (a well-known Mpro inhibitor) was determined. As shown in Figure 3, the A_585_/A_520_ value was intensified and the solution turned purple when the Mpro was incubated with increasing concentrations of ebselen, demonstrating that higher concentrations of ebselen could inhibit the Mpro activity more effectively. The half-maximum inhibition value (IC_50_) was calculated according to the inhibition rate (%), described by the following equation:$$\mathrm{inhibition}\;\mathrm{rate}\;(\%)\;=\frac{A^{1}-A}{A^{1}-A^{0}}\;\times \%$$
where A^0^ and A^1^ represent the A_585_/A_520_ in the absence and presence of the Mpro at a fixed concentration; A represents the A_585_/A_520_ in the presence of the Mpro and a given concentration of ebselen. Based on the relationship between the inhibitor rate and the ebselen concentration, the IC_50_ was found to be 7.6 nM, which is consistent with that obtained by the fluorescence kit. Thus, the colorimetric method can be employed to screen the potential Mpro inhibitors for the development of novel antiviral drugs.

### 2.3. Feasibility for Electrochemical Detection of Mpro

Although the colorimetric method is simple and easy to operate, other components in biological samples may adsorb onto the surface of bare AuNPs [31], thus limiting its application for the determination of low concentrations of Mpro in biological matrixes. Based on the principle of the peptide-induced aggregation of AuNPs, we propose that AuNPs could be assembled on the peptide-covered electrode surface due to the similar environment of the gold electrode and AuNPs. The detection principle based on the co-assembly of peptide and AuNPs are illustrated in Figure 1B. The peptide anchored on the electrode can capture AuNPs via the electrostatic interactions. Then, the captured AuNPs recruit the peptides (RKTSAVLQSGFRK) in the solution via the same interactions. The repeated recruitment of AuNPs and peptide will lead to the formation of peptide/AuNPs aggregates on the electrode surface, thus facilitating the electron transfer of [Fe(CN)_6_]^3−/4−^ due to the excellent conductivity and large specific surface area of AuNPs. Once the peptide-covered electrode was pre-incubated with the Mpro, the peptide on the electrode surface would be specifically recognized and cleaved, thus limiting the attachment of AuNPs and preventing the co-assembly of AuNPs and peptide on the electrode surface. As shown in Figure 4A, the electron transfer resistance (R_et_) of the peptide-covered electrode (curve one) decreased after incubation with the mixture of AuNPs and peptide (curve two) or AuNPs (curve three). The R_et_ for the mixture is greatly lower than that for the AuNPs alone, and no significant change for the R_et_ was observed when the peptide-covered electrode was incubated with the peptide only (data not shown). The result indicated the decrease in the R_et_ (curve two) should be attributed to the co-assembly of AuNPs and peptide. When the peptide-covered electrode was treated by the Mpro, the R_et_ was intensified slightly (curve four), demonstrating that the direct EIS detection of the Mpro by the enzymatic hydrolysis of the peptide substrate on the electrode surface was less sensitive. When the Mpro-treated electrode was incubated with the mixture of AuNPs and peptide, no significant change in the R_et_ was observed (curve five). The result indicated that the cleavage of the peptide prevented the co-assembly of AuNPs and peptide on the electrode surface. The impedance change (ΔR_et_) between curve two and curve five is greater than that between curve one and curve four, indicating that the sensitivity can be improved by simply treating the sensor electrode with the mixture of AuNPs and peptide. The results were also confirmed by cyclic voltammogram (CV). As depicted in Figure 4B, the incubation of the peptide-covered electrode (curve one) with AuNPs/peptide (curve two) or AuNPs alone (curve three) led to the increase of peak currents, confirming the capture and formation of the peptide/AuNPs networks. When the peptide-covered electrode was treated by the Mpro, the current decreased slightly (cf. curve one and curve four). No significant change was observed when the Mpro-treated sensing electrode was incubated with AuNPs/peptide (cf. curve four and curve five), indicating that the cleavage of the peptide substrate prevented the attachment of AuNPs and the follow-up formation of the AuNPs/peptide networks.

### 2.4. Sensitivity and Selectivity of the Electrochemical Method

The sensitivity of the electrochemical method was investigated by analyzing the different concentrations of the Mpro. As depicted in Figure 5, the R_et_ increased correspondingly with the increase of the Mpro concentration. A linear relationship was attained between the average ΔR_et_ value of three trails and the Mpro concentration in the range of 0.1~15 pM. The linear equation was found to be ΔR_et_ = 448.9 [Mpro] (pM) + 315.5 with a detection limit down to 0.1 pM. The value is lower than that of the above colorimetric method, demonstrating that the electrochemical method showed relatively higher sensitivity.

The selectivity of the method was evaluated by testing BSA, IgG, thrombin and PKA. As shown in Figure 6, the ΔR_et_ value for the Mpro was significantly higher than that for the four proteins, even at a high concentration, demonstrating that the method exhibits excellent selectivity. Additionally, the anti-interference ability of the method was investigated by determining the Mpro in 10% serum. As a result, the ΔR_et_ for determining the Mpro in the serum is not significantly different from that in the buffer. This suggests that the electrochemical method exhibits great potential in biological sample analysis and clinical investigation.

## 3. Materials and Methods

### 3.1. Chemicals and Reagents

Mpro kits were purchased from Beyotime Biotechnology (Shanghai, China). 11-Mercapto-1-undecanol (MUA), tris(2-carboxyethyl)phosphine (TCEP), bovine serum albumin (BSA), IgG, thrombin, protein kinase (PKA) and serum were purchased from Sigma-Aldrich (Shanghai, China). Chloroauric (III) acid, citrate, phosphate and other reagents were obtained from Aladdin Biochemical Technology Co., Ltd. (Shanghai, China). Peptides were prepared by the solid–solid phase synthesis method with a Focus XI peptide synthesizer.

### 3.2. Peptide-Triggered Aggregation of AuNPs

AuNPs with an average size of 13 nm were prepared through the citrate reduction method. The prepared AuNPs were diluted with 10 mM phosphate (pH 7.0) to a given concentration. Then, 50 μL of AuNPs suspension was added to 50 μL of peptide (RKTSAVLQSGFRK) at different concentrations. After incubation for 5 min, the change of solution color was observed by eyes and the UV-Vis spectra were collected on a Cary 50 spectrophotometer. The aggregation of AuNPs was confirmed by dynamic light scattering (DLS) analysis performed on a Zeta Sizer Nano ZS90 (Malvern Company, Worcestershire, England).

### 3.3. Colorimetric Assays of Mpro Activity

Mpro was diluted with phosphate buffer to different concentrations. Then, 25 μL of Mpro was mixed with 25 μL of RKTSAVLQSGFRK peptide in a centrifugal tube. After reaction at 37 °C for 30 min, 50 μL of AuNPs suspension was added to the mixture for 5 min incubation. After that, the color change was observed and the UV-Vis spectra were collected. For the inhibition analysis, 20 μL of 1 nM Mpro was first mixed with 5 µL of ebselen at different concentrations. After incubation for 10 min to inhibit the Mpro activity, 25 μL of peptide was added to the Mpro/ebselen mixed solution. The other procedures are similar to those for monitoring the activity of pure Mpro sample.

### 3.4. Preparation of Sensing Electrode

The gold electrode was incubated with piranha solution, rinsed with water and polished with 0.05 μm aluminum powder. After being washed in 50% ethanol under sonication, the electrode was electrochemically scanned in 0.5 M sulfuric acid. After being rinsed with water and dried with nitrogen, the cleaned electrode was immersed in the peptide solution (10 μM CPPPPKTSAVLQSGFRK and 0.5 mM TCEP in phosphate buffer) at 4 °C overnight. After that, the electrode was incubated with 10 mM MUA for 1 h to block the unreacted gold surface. Finally, the peptide-covered electrode was rinsed with 50% ethanol and used for the detection of Mpro. In this work, the CPPPP segment is an effective linker and spacer for peptide immobilization to attain high cleavage efficiency [32,33].

### 3.5. Electrochemical Detection of Mpro

For the electrochemical analysis of Mpro, the peptide-covered electrode was incubated with Mpro at a given concentration for 30 min to allow the hydrolysis of the peptide. Then, the electrode was incubated with 25 μL of AuNPs suspension, followed by the addition of 25 μL of RKTSAVLQSGFRK peptide. After being gently rinsed with water, the electrode was placed in 5 mM [Fe(CN)_6_]^3−/4−^ for EIS measurement.

## 4. Conclusions

In summary, this work indicated that the peptide substrate of the Mpro could induce the aggregation and color change of AuNPs. Cleavage of the peptide by Mpro prevented the aggregation of AuNPs and thus facilitated the development of a colorimetric method for the analysis of Mpro activity and the screening of its inhibitors. In contrast to the commercial fluorescence kit, the method features the advantages of being low cost, having good stability, facile operation and naked-eye readout. Furthermore, we developed a sensitive surface-tethered method for Mpro detection by implanting the principle of the colorimetric assay into the electrochemical field. In contrast to the colorimetric method, the electrochemical method showed a higher sensitivity and a better anti-interference ability and has been used for the assay of Mpro in a 10% human serum with a satisfactory result. The proposed approach based on the co-assembly of AuNPs and peptide on the electrode surface should be promising as a general strategy for the design of protease biosensors by matching the sequence of the peptide substrate and using appropriate types of nanomaterials.

## Data Availability

Not applicable.

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
