# Peer review of "Colorimetric and Electrochemical Methods for the Detection of SARS-CoV-2 Main Protease by Peptide-Triggered Assembly of Gold Nanoparticles"

_molecules, 2022, doi:10.3390/molecules27030615_

Round 1

Reviewer 1 Report

In proposed study authors present sensing technique for detection of SARS-CoV-2 specific main protease (Mpro) on the basis of agglumoration of AuNPs via colorimetric and electrochemical sensing. Proposed study has innovative concept, but it is content very limited and unjustified data. I suggest to reject the manuscript and suggest authors to modify their manuscript after introducing following comments before submitting in another Journal:

  1. What is the basis of using two different peptides for colorimetric and electrochemical sensing when target is same? Or these peptides were going to bind with AuNPs?
  2. How authors calculated the half-maximal inhibitory concentration (IC50) of ebselen to be 7.6 nM?
  3. Electrochemical study does not only dependent on EIS. I suggest authors to introduce complimentary CV or DPV data of electrode fabrication. As per EIS data-points in this study (Fig. 4) exhibits higher conductivity of AuNPs/peptide in comprision to AuNPs alone, which further decreased in presence of Mpro. Generally, biological components such as peptides are non-conducting and justify the high Ret values which is reverse in this study. Additionally, interaction of Mpro with peptide must have generated more electron in comparision to AuNPs/peptides, then why curves 4 and 5 have high impedance? Justify your results in details with potential biochemical equations.
  4. Mpro concentration is different in Fig. 5(A) with values written in text of sub-section 3.4. Which target range is correct? And what are these different target concentrations, only range without any difference is not enough. Mention each concentration.
  5. The values of ΔRet (ΔRet= Ret Mpro n (n=0.1…… 50 pM) - Ret Mpro conc.n (n=0))in Fig. 5(B) is not justified via Fig. 4 and 5(A). For example, 4th curve (cyan blue color) in Fig. 5(A) has Ret value of 4500 ohm and Ret at 0 Mpro concentration is ~1200 ohm. Now, if we see 4th data point of Fig. 5(B), ΔRet value is ~4500 which should be ~3300 ohm. Explain.
  6. Optimization of developed sensor is missing from the manuscript. Please incorporate precession, accuracy, reproducibility, and analytical performance of developed sensor in clinical samples or spiked clinical samples.
  7. Add following references at appropriate places in manuscript: https://doi.org/10.1080/14737159.2020.1816466, https://doi.org/10.1002/jmv.27358 

Author Response

We thank the reviewer for his/her comments: “In proposed study authors present sensing technique for detection of SARS-CoV-2 specific main protease (Mpro) on the basis of agglumoration of AuNPs via colorimetric and electrochemical sensing. Proposed study has innovative concept, but it is content very limited and unjustified data. I suggest to reject the manuscript and suggest authors to modify their manuscript after introducing following comments before submitting in another Journal:

Comment 1: What is the basis of using two different peptides for colorimetric and electrochemical sensing when target is same? Or these peptides were going to bind with AuNPs?

Response: It is a good question. In the colorimetric assays, the label-free peptide with a sequence of RKTSAVLQSGFRK was used to trigger the aggregation of AuNPs through the electrostatic interactions. Cleavage of the peptide by Mpro inhibited the aggregation of AuNPs. In the electrochemical assays, the peptide with a sequence of CPPPPKTSAVLQSGFRK was immobilized on the electrode surface through the formation of Au-S bond. The PPPP segment between the substrate and Cys residue can make the peptide protrude outward from the electrode surface so that a higher cleavage efficiency can be attained (J. Am. Chem. Soc., 2012, 134 6000–6005; Sens. Actuat. B: Chem., 2020, 320, 128436). Cleavage of the peptide by Mpro made it lose the ability to capture AuNPs through the electrostatic interaction. Thus, the assembly of AuNPs on the electrode surface was prevented.

Comment 2: How authors calculated the half-maximal inhibitory concentration (IC50) of ebselen to be 7.6 nM?

Response: The half-maximum inhibition value (IC50) was calculated according to the inhibition rate (%) described by the following equation:

where A0 and A1 represent the A585/A520 in the absence and presence of Mpro at a fixed concentration; A represents the A585/A520 in the presence of Mpro and a given concentration of ebselen. The IC50 was determined based on the relationship between inhibitor rate and ebselen concentration. We have added the sentences in the revised text.

Comment 3: Electrochemical study does not only dependent on EIS. I suggest authors to introduce complimentary CV or DPV data of electrode fabrication. As per EIS data-points in this study (Fig. 4) exhibits higher conductivity of AuNPs/peptide in comprision to AuNPs alone, which further decreased in presence of Mpro. Generally, biological components such as peptides are non-conducting and justify the high Ret values which is reverse in this study. Additionally, interaction of Mpro with peptide must have generated more electron in comparision to AuNPs/peptides, then why curves 4 and 5 have high impedance? Justify your results in details with potential biochemical equations.

Response: The peptide/AuNPs networks can promote the electron transfer of [Fe(CN)6]3−/4 due to the excellent conductivity and large specific surface area of AuNPs (Sens. Actuat. B: Chem. 2017, 239, 834). Moreover, the positive charges of peptide framework can also promote the electron transfer of [Fe(CN)6]3−/4 through the electrostatic interaction (Biosens. Bioelectron. 2013, 45, 1–5). The results of cyclic voltammogram (CV) have been added in Fig. 4B. We thank the reviewer for his/her suggestion.

Comment 4: Mpro concentration is different in Fig. 5(A) with values written in text of sub-section 3.4. Which target range is correct? And what are these different target concentrations, only range without any difference is not enough. Mention each concentration.

Response: We have revised the spelling mistake. The linear relationship was attained in the range of 0.1 ~ 15 pM. The concentrations for the tested samples were added in the figure caption.

Comment 5: The values of ΔRet (ΔRet= Ret Mpro n (n=0.1…… 50 pM) - Ret Mpro conc.n (n=0)) in Fig. 5(B) is not justified via Fig. 4 and 5(A). For example, 4th curve (cyan blue color) in Fig. 5(A) has Ret value of 4500 ohm and Ret at 0 Mpro concentration is ~1200 ohm. Now, if we see 4th data point of Fig. 5(B), ΔRet value is ~4500 which should be ~3300 ohm. Explain.

Response: The Ret value in the absence of Mpro (black curve) is 1274 ohm and that in the presence of 5 pM Mpro (cyan blue curve) was 4340 ohm. The ΔRet was 3066 ohm. In Figure 5B, the data was 2920 ohm, which is an average value of three trails. We have explained it in the revised text.

Comment 6: Optimization of developed sensor is missing from the manuscript. Please incorporate precession, accuracy, reproducibility, and analytical performance of developed sensor in clinical samples or spiked clinical samples.

Response: The analytical performances have been expressed by the linear range, detection limit or lowest detectable concentration, selectivity and relative standard deviations (RSDs, shown as the error bars) for the assays of different samples. The information has been discussed in the text.

Comment 7: Add following references at appropriate places in manuscript: https://doi.org/10.1080/14737159.2020.1816466, https://doi.org/10.1002/jmv.27358.

Response: We have added the references in the revised manuscript.

Reviewer 2 Report

The research about SARS-CoV-2 is a “hot” topic. In this work, the authors reported two methods for the detection of SARS-CoV-2 main protease Mpro and screening of its inhibitors. The colorimetric method is very simple and the electrochemical method is more sensitive. The manuscript is well written and the results are acceptable. I recommend it for publication after minor revision.

  1. Why CPPPPKTSAVLQSGFRK but not CKTSAVLQSGFRK was used as the substrate of electrochemical assays?
  2. The experimental conditions should be provided in the figure caption or the experimental part, such as the concentration of BSA, IgG, thrombin, PKA and Mpro in Figure 6.
  3. The references should be updated. More references about peptide-induced aggregation of AuNPs should be cited.
  4. How and why the peptide/AuNPs networks can promote the electron transfer of [Fe(CN)6]3−/4?

Author Response

We thank the reviewer for his/her positive and constructive comments: “The research about SARS-CoV-2 is a “hot” topic. In this work, the authors reported two methods for the detection of SARS-CoV-2 main protease Mpro and screening of its inhibitors. The colorimetric method is very simple and the electrochemical method is more sensitive. The manuscript is well written and the results are acceptable. I recommend it for publication after minor revision.

Comment 1: Why CPPPPKTSAVLQSGFRK but not CKTSAVLQSGFRK was used as the substrate of electrochemical assays?

Response: It is a good question. In the electrochemical assays, the peptide with a sequence of CPPPPKTSAVLQSGFRK was immobilized on the electrode surface through the formation of Au-S bond. The PPPP segment between the substrate and Cys residue can make the peptide protrude outward from the electrode surface so that higher cleavage efficiency can be attained (J. Am. Chem. Soc., 2012, 134 6000–6005; Sens. Actuat. B: Chem., 2020, 320, 128436). We have explained it in Part 2.4.

Comment 2: The experimental conditions should be provided in the figure caption or the experimental part, such as the concentration of BSA, IgG, thrombin, PKA and Mpro in Figure 6.

Response: The concentrations for the tested samples have been added in the figure caption.

Comment 3: The references should be updated. More references about peptide-induced aggregation of AuNPs should be cited.

Response: More references have been added in the revised manuscript.

Comment 4: How and why the peptide/AuNPs networks can promote the electron transfer of [Fe(CN)6]3−/4?

Response: The peptide/AuNPs networks can promote the electron transfer of [Fe(CN)6]3−/4 due to the excellent conductivity and large specific surface area of AuNPs and the positive charges of peptide framework (Sens. Actuat. B: Chem. 2017, 239, 834; Biosens. Bioelectron. 2013, 45, 1–5). We have written the sentence in Introduction.